# Reliability and Validity of the Dyskinesia Impairment Scale in Children and Young Adults with Inherited or Idiopathic Dystonia

**DOI:** 10.3390/jcm9082597

**Published:** 2020-08-11

**Authors:** Annika Danielsson, Inti Vanmechelen, Cecilia Lidbeck, Lena Krumlinde-Sundholm, Els Ortibus, Elegast Monbaliu, Kristina Tedroff

**Affiliations:** 1Department of Women’s and Children´s Health, Karolinska Institutet, 17176 Stockholm, Sweden; cecila.lidbeck@ki.se (C.L.); lena.krumlinde.sundholm@ki.se (L.K.-S.); kristina.tedroff@ki.se (K.T.); 2Sach’s Children and Youth Hospital, Stockholm South General Hospital, 11883 Stockholm, Sweden; 3Department of Rehabilitation Sciences, KU Leuven Campus Bruges, 8200 Bruges, Belgium; inti.vanmechelen@kuleuven.be (I.V.); elegast.monbaliu@kuleuven.be (E.M.); 4Neuropediatric Department, Karolinska University Hospital, 17176 Stockholm, Sweden; 5Department of Development and Regeneration, KU Leuven, 3000 Leuven, Belgium; els.ortibus@uzleuven.be; 6Cerebral Palsy Clinic, University Hospitals Leuven-Campus Pellenberg, 3000 Leuven, Belgium; 7Centre for Developmental Disabilities, 3000 Leuven, Belgium

**Keywords:** Dyskinesia Impairment Scale, Burke–Fahn–Marsden Dystonia Rating Scale, dystonia, choreoathetosis, assessment, reliability, validity

## Abstract

Background: The Dyskinesia Impairment Scale (DIS) is a new assessment scale for dystonia and choreoathetosis in children and youth with dyskinetic cerebral palsy. Today, the Burke–Fahn–Marsden Dystonia Rating Scale (BFM) is mostly used to assess dystonia in children with inherited dystonia. The aim of this study was to assess reliability and validity of the DIS in children and youth with inherited or idiopathic dystonia. Methods: Reliability was measured by (1) the intraclass correlation coefficients (ICCs) for inter-rater and test-retest reliability, as well as (2) standard error of measurement (SEM) and minimal detectable difference (MDD). For concurrent validity of the DIS-dystonia subscale, the BFM was administered. Results: In total, 11 males and 9 females (median age 16 years and 7 months, range 6 to 24 years) were included. For inter-rater reliability, the ICCs for the DIS total score and the dystonia and choreoathetosis subscale scores were 0.83, 0.87, and 0.71, respectively. For test-retest reliability, the ICCs for the DIS total score and the dystonia and choreoathetosis subscale scores were 0.95, 0.88, and 0.93, respectively. The SEM and MDD for the total DIS were 3.98% and 11.04%, respectively. The Spearman correlation coefficient between the dystonia subscale and the BFM was 0.88 (*p* < 0.01). Conclusions: Good to excellent inter-rater, test-retest reliability, and validity were found for the total DIS and the dystonia subscale. The choreoathetosis subscale showed moderate inter-rater reliability and excellent test-retest reliability. The DIS may be a promising tool to assess dystonia and choreoathetosis in children and young adults with inherited or idiopathic dystonia.

## 1. Introduction

Dystonia is defined as a movement disorder characterized by sustained or intermittent muscle contractions causing abnormal, often repetitive, movements, postures, or both [1]. It is a heterogeneous condition, with varying age of onset, etiology and clinical features that range from focal dystonia to severe generalized dystonia [1]. Inherited dystonia is defined as dystonia forms of proven genetic origin, whereas idiopathic dystonia is dystonia with an unknown cause [1].

Paediatric-onset dystonia is a debilitating condition that negatively impacts the quality of life of children [2]. The dystonia is mostly generalized and often combined with other movement disorders, such as choreoathetosis, spasticity, myoclonus, or ataxia [3]. Although multiple treatment options are available for generalized dystonia, the clinical effect is often insufficient and the current scientific evidence for treatment management is scarce [2,4].

A reliable measurement of dystonia is crucial, in the clinical setting as well as in the research setting, in order to evaluate the effect of interventions and to assess the disease course. Today, there are a handful of rating scales for generalized dystonia syndromes [5,6,7,8,9]. The majority of these scales are primarily developed for adults and their application for children might yield incorrect results, partly since typical motor development is not accounted for [10]. There are to date, however, no rating scales designed to assess dystonia in children with inherited dystonia [11]. This lack of reliable assessment hinders the correct interpretation of the effect of intervention studies.

The rater is a crucial “part” of the measuring system in clinically used dystonia rating scales. In the case of dystonia rating scales, the rater must observe movements or rest postures and apply operational criteria to subjective observations, thus imposing a varying degree of subjectivity [12]. The measure of inter-rater reliability can be used to assess whether different raters are in agreement about the variable being assessed. Another indispensable characteristic of a rating scale is its capacity to produce stable measures over time (i.e., with repeated administration of the test). For this purpose, test-retest reliability can be calculated [12]. Validity of an instrument, on the other hand, concerns the extent to which an instrument measures what it is intended to measure. To evaluate a new measure against a gold standard, concurrent validity can be calculated [12].

The Burke–Fahn–Marsden Dystonia Rating Scale (BFM) was developed to assess generalized inherited dystonia in adults and was the only scale recommended to evaluate generalized dystonia in a recent systematic review [6,11]. Due to the lack of reliable and valid assessment scales for the pediatric population, the BFM has often been used in intervention studies in children. However, the authors themselves recommend caution when interpreting the results of the BFM in intervention studies for other forms of dystonia or other target populations [10,11]. Additionally, the sensitivity of the scale has been questioned due to the lack of distinction between distal and proximal limbs when scoring dystonia [11,13].

The Dyskinesia Impairment Scale (DIS) was recently developed to evaluate dystonia and choreoathetosis in children and young adults with dyskinetic cerebral palsy (DCP) [5]. Choreoathetosis is characterized as hyperkinesia and muscle tone fluctuation and is clinically distinct from dystonia [3]. The DIS addresses the need for a dystonia rating scale including the assessment of choreoathetosis, as these movement disorders often co-occur in DCP.

Compared to the BFM, the DIS has some advantages, the most obvious one being that the DIS assesses both dystonia and choreoathetosis, but there are other important aspects. Firstly, the DIS is based upon most recent definitions for dystonia and choreoathetosis, leading to a more accurate description of these movement disorders. Secondly, the scale discriminates between amplitude and duration of each movement disorder, in action and at rest, and between distal and proximal distributions of the limbs [5]. Thirdly, the DIS evaluates only the presence and severity of dystonia and choreoathetosis, whereas the BFM is constructed in such a way that the impact of dystonia on function is, by default, evaluated along with the severity of dystonia, which may complicate the interpretation of the results. Finally, the DIS has showed high inter-rater and test-retest reliability and validity in DCP [14,15].

The current study aims to examine the inter-rater and test-retest reliability and concurrent validity of the DIS in children and young adults with inherited or idiopathic dystonia. If the DIS scores are found reliable to assess dystonia and choreoathetosis in this population, the movement disorders can be better understood, followed, and, possibly, more adequately treated.

## 2. Experimental Section

This is a cross-sectional, two-center study, evaluating an assessment, the DIS, for dystonia and choreoathetosis. The BFM was applied to calculate concurrent validity of the dystonia subscale of the DIS. Ethical approval was obtained from the Regional Ethics Committee in Stockholm, Sweden, (no 2016/682-31/2) and the Ethical Committee of the KU Leuven, Belgium (S60866). Written informed consent was obtained from all participants or, when appropriate, their caregivers. The study adhered to the Declaration of Helsinki.

### 2.1. Participants

The required sample size was calculated according to the methods of Bujang et al. [16]. With the null hypothesis being that the intra-class correlation coefficient (ICC) values would be >0.7 and the alternative hypothesis that the ICC values would be >0.9, alpha being 0.05 and with power set to 90%, at least 18 participants should be included. Our sample size was therefore set to 20 participants. Participants were recruited from outpatient neuropediatric clinics from Astrid Lindgren Children’s Hospital and Sachs’ Children and Youth Hospital in Stockholm, Sweden, and from the University Hospitals of KU Leuven, Belgium. Inclusion criteria were individuals 5–25 years of age with a diagnosis of inherited or idiopathic dystonia obtained by a pediatric neurologist. Exclusion criteria were (1) paroxysmal diseases and (2) dopa-responsive dystonias.

### 2.2. Assessments

The DIS assesses dystonia and choreoathetosis (Figure 1) and consists of one DIS total score and two subscale scores, one for dystonia (DIS-D) and one for choreoathetosis (DIS-CA). Each subscale evaluates the presence of dystonic and choreoathetotic movements in 12 body regions, including the eyes, mouth, neck, trunk, and limbs, during two actions and one rest posture each. For the limbs, a distinction is made between the proximal and distal parts and between the right and the left sides. For each body region, the amount of time that the dystonia and choreoathetosis is present (duration) and the range of motion of the dyskinetic movement (amplitude) are scored separately on a 5-pointscale from 0 to 4, where lower scores indicate lower presence and severity of dystonia and choreoathetosis, respectively. As a baseline for the amplitude scores of the DIS, passive range of motion of the extremities are measured with a goniometer [17]. The scores in each subscale range from 0 to 288, including action scores (ranging from 0 to 192), and rest scores (ranging from 0 to 96). The DIS total score is the sum of the dystonia and choreoathetosis subscale scores (ranging from 0 to 576). If an action is impossible to perform for the participant, that item is removed and the total DIS percentage is recalculated without this item. The DIS total score and subscale scores are therefore presented as a percentage of the maximum score of the scale. The DIS is based upon a standardized video protocol containing all postulated rest postures and actions (Appendix A). The scoring requires specific training, which is presently available through courses at KU Leuven, Belgium.

The BFM assesses dystonia and consists of two subscales, one movement scale (BFM-M), based on observation of the patient, and one disability scale (BFM-D), based on the patients or caregivers report of disability in activities of daily living. In the present study, only the BFM-M was used to assess dystonia. The BFM-M rates dystonia severity and provoking factors in nine body areas, including the eyes, mouth, speech/swallowing, neck, trunk, and arms/legs on left and right sides, on a 5-point scale from 0 to 4. The provoking factor refers to the situation in which the dystonia occurs, where 0 = no dystonia at rest or with action and 4 = dystonia at rest. The score obtained for the eyes, mouth, and neck are each multiplied with 0.5, before being entered into the calculation of the total score, in order to down-weight them. The total BFM-M subscore is provided by the sum of the products of the provoking, severity, and weighing factors. The maximum BFM-M score is 120, with a lower score indicating less severe dystonia. In this study, the item speech and swallowing in the BFM-M was excluded, since swallowing is not scored through observation, but via caregiver records, thus giving a maximum score of 112.

Functional ability was classified using the Gross Motor Function Classification System-Expanded and Revised (GMFCS-ER) and the Manual Ability Classification System (MACS), describing gross motor function and the ability to handle objects in daily activities, respectively, in children and youth with CP [18,19,20]. Although designed for the population with CP, these classification systems are used in this study due to the lack of classifications describing motor function in inherited or idiopathic dystonia.

### 2.3. Procedure

All participants were invited to attend two visits to be video recorded according to the DIS video protocol, with a maximum of 2 weeks between visits. Participants from Belgium were video recorded in their habitual environment (school and home) and patients from Stockholm were video recorded at the Motion Analysis Laboratory at Astrid Lindgren Children’s Hospital. The videos were used to score both the DIS and the BFM-M.

In Stockholm, Sweden, demographic information regarding age at disease onset and functional gross motor and manual abilities was collected during the first visit. In Leuven, Belgium, this information was gathered from medical records after the visits. Information regarding genetic aberrations was obtained from medical records.

### 2.4. Scoring Process

All videos were independently scored by the same three raters within a 4 week time frame. Raters included a pediatric neurologist, a physiotherapist and a movement scientist, and all scoring was done in Sweden. The raters had attended a DIS instructional course in Belgium and had clinical experience in dystonia. Before scoring the participants in the study, the three raters scored four training videos together, according to the DIS and BFM-M. Any issues/doubts were at this point discussed with the developer (EM) of the DIS who is experienced in scoring the BFM-M. Scoring sheets, definitions of dystonia and choreoathetosis, and body region descriptions of the DIS and the BFM-M can be found in Appendix A.

Video montages compatible with the scoring order of the DIS were made from all original videos in order to simplify the scoring process. For evaluation of inter-rater reliability and concurrent validity, videos from visit 1 were independently scored according to the DIS and the BFM-M scale by all three raters. The videos were scored in different orders by the raters. For test-retest evaluation, the videos from visit 2 were evenly distributed between the three raters, and each video was scored according to the DIS by one rater. The DIS outcome from visit 2 was paired up with the same scorer’s ratings of visit 1 to evaluate test-retest reliability.

### 2.5. Statistical Analysis

For inter-rater reliability, the two way random effects model (single rater, absolute agreement) was selected for calculation of the ICC values since: (1) each subject is assessed by the same set of raters; (2) generalization of the raters towards a population of raters with the same characteristics is possible; (3) when using the DIS in clinical practice, a single rater scores the videos from one assessment; and (4) absolute agreement and not consistency is of interest. For test-retest reliability, the two way mixed effects model (single rater, absolute agreement) was used, as there was no intention to generalize the findings beyond the raters involved. The rater is therefore a fixed effect, and the subject is a random effect.

In accordance with Portney and Watkins, ICC higher than 0.90 was interpreted as excellent, an ICC between 0.75 and 0.90 as good, between 0.60 and 0.75 as moderate and <0.60 as poor [12]. The standard error of measurement (SEM) and minimal detectable difference (MDD) for test-retest was calculated for the total DIS, the DIS dystonia and choreoathetosis subscales using the formulae [SEM = SD × √(1−ICC)] and [MDD = SEM × 1.96 × √2]. SEM and MDD are presented here as percentage points, since the total DIS score and total subscale scores are presented as percentages of the maximum values of the total DIS score and total subscale scores, respectively. Concurrent validity was determined by Spearman correlation coefficient and calculated as the correlation between the scores of the BFM-M and the scores of the DIS-D. All statistics were calculated with SPSS version 25 (SPSS Inc., Chicago, IL, USA).

## 3. Results

In total, 20 individuals, 11 males and 9 females, with inherited or idiopathic dystonia (median age of 16 years and 7 months, range 6 years and 5 months to 24 years and 1 month) were included in the study, 6 from Belgium and 14 from Stockholm. All completed the first visit and 19 individuals came to a second visit within a time frame ranging from a minimum of 2 hours to maximum 2 days between visits. One patient was not invited to the second visit, since she was considered by the investigators to be uncomfortable during the first visit. Patient characteristics and genetic information are shown in Appendix A.

The GMFCS-ER level was evenly distributed among the participants: GMFCS-ER I (*n* = 3), GMFCS-ER II (*n* = 4), GMFCS-ER III (*n* = 3), GMFCS-ER IV (*n* = 5), and GMFCS-ER V (*n* = 5). The MACS level showed a similar pattern: MACS I (*n* = 3), MACS II (*n* = 4), MACS III (*n* = 4), MACS IV (*n* = 2) and MACS V (*n* = 8).

For the rest postures, all participants were scored for all body regions. For the action scores, participants who were unable to execute the requested action did not receive a score for this particular action. Since 10 out of the 20 participants were at GMCS-ER level IV or V, they were unable to stand or walk. This led to missing values for the body region ‘leg proximal’, which is scored during active standing. Additionally, 10 out of 20 participants were unable to perform the requested actions to score the trunk region, which are forward bending and active sitting. The resulting scores of these actions are thus not based on 20 participants, but only on the number of participants able to execute the requested action.

### 3.1. Inter-Rater Reliability

The ICCs for the total DIS, DIS-D, and DIS-CA scores and the following subcategories for action and rest and duration and amplitude are shown in Table 1. The total score of the DIS showed good inter-rater reliability, with an ICC of 0.83. In addition, the DIS-D score showed good inter-rater reliability, with an ICC of 0.87. The DIS-CA score showed moderate inter-rater reliability with an ICC of 0.71.

#### 3.1.1. DIS-D, Action, and Rest

During action, good inter-rater reliability was found with an ICC of 0.85. For action duration and action amplitude, inter-rater reliability was good with ICC 0.77 and 0.85, respectively. For the body regions, the ICCs ranged from poor to moderate for action duration, action amplitude, and the sum of action amplitude and duration.

During rest, good inter-rater reliability was found with an ICC of 0.82. For rest duration and rest amplitude, inter-rater reliability was good with ICC 0.76 and 0.82, respectively. For the body regions, ICCs were poor to moderate for rest duration, rest amplitude and the sum of rest amplitude and duration, with the exception of the proximal legs, which showed good ICCs for rest amplitude and the sum of action duration and amplitude.

#### 3.1.2. DIS-CA, Action, and Rest

During action, moderate inter-rater reliability was found with an ICC of 0.61. For action duration and action amplitude, inter-rater reliability was moderate with ICC 0.61 and 0.60, respectively. In the body regions, the ICCs ranged from poor to moderate for action duration, action amplitude and the sum of action amplitude and duration.

During rest, good inter-rater reliability was found with an ICC of 0.79. For rest duration and rest amplitude, inter-rater reliability was good with ICC 0.81 and 0.78, respectively. For the body regions, ICCs were poor to good for rest duration, poor to moderate for rest amplitude and poor to good for the sum of rest amplitude and duration.

### 3.2. Test-Retest Reliability

The ICCs for the total DIS, DIS-D, and DIS-CA scores and the following subcategories for action and rest and duration and amplitude are shown in Table 2. The total score of the DIS showed excellent test-retest reliability with an ICC of 0.95. For DIS-D, the ICC scores showed good test-retest reliability with an ICC of 0.88. For DIS-CA, the ICC scores showed excellent test-retest reliability with an ICC of 0.93.

#### 3.2.1. DIS-D, Action, and Rest

During action, good test-retest reliability was found with an ICC of 0.88. For action duration and action amplitude, test-retest reliability was excellent with ICC 0.90 and 0.92, respectively. In the body regions, the ICCs ranged from poor to good for action duration and action amplitude and from poor to excellent for the sum of action amplitude and duration.

During rest, good test-retest reliability was found with an ICC of 0.79. For rest duration and rest amplitude, test-retest reliability was good with ICC 0.76 and 0.77, respectively. For the body regions, ICCs were poor to good for rest duration, rest amplitude and the sum of rest amplitude and duration.

#### 3.2.2. DIS-CA, Action, and Rest

During action, excellent test-retest reliability was found with an ICC of 0.91. For action duration and action amplitude, test-retest reliability was excellent with ICC 0.83 and 0.87, respectively. In the body regions, the ICCs ranged from poor to moderate for action duration, from poor to good for action amplitude and from poor to moderate for the sum of action amplitude and duration.

During rest, good test-retest reliability was found with an ICC of 0.86. For rest duration, test-retest reliability was good with an ICC of 0.83, and for rest amplitude, test-retest reliability was excellent with an ICC of 0.93. For the body regions, ICCs were poor to good for rest duration, for rest amplitude and for the sum of rest amplitude and duration.

### 3.3. Standard Error of Measurement and Minimal Detectable Difference

The SEM and MDD for the total DIS were 3.98% and 11.04%, respectively. For the DIS-D, the SEM and MDD were 8.16% and 22.61%, respectively. For the DIS-CA, the corresponding SEM and MDD were 3.80% and 10.54%, respectively.

### 3.4. Concurrent Validity

The Spearman correlation coefficient between the DIS-D and BFM-M was 0.88 (*p* < 0.01).

## 4. Discussion

This study found good inter-rater reliability for the total DIS and the DIS-D and moderate inter-rater reliability for the DIS-CA. Furthermore, we found excellent test-retest reliability for the total DIS and for the DIS-CA, and good test-retest reliability for the DIS-D. This high test-retest reliability also indicates high intra-rater reliability, although this was not formally assessed in this study. Concurrent validity of the DIS-D, when compared with the gold standard BFM-M, was good. Altogether, these results show that the DIS might be a valuable rating scale for dystonia and choreoathetosis in children and young adults with inherited or idiopathic dystonia.

Inter-rater reliability for the total DIS and the DIS-D in the present study is slightly lower than the original DIS study for DCP, but in agreement with previous results from another reliability study of the DIS with raters inexperienced in assessing dystonia but specifically trained in the DIS [14]. For the DIS-CA, all total values and body regions for both action and rest were lower than in previous studies in DCP [5,14]. For the DIS-D, inter-rater reliability for total values and body regions during action was in agreement with the original DIS study whereas during rest, inter-rater reliability was lower than in the original DIS study, although higher than in the study with inexperienced raters [5,14]. These findings indicate that the DIS is as reliable in inherited and idiopathic dystonia as in DCP regarding dystonia assessment. With regard to choreoathetosis, our findings showed a lower reliability for choreoathetosis compared with the DCP population. Therefore, in depth-analysis of raw data was performed in order to understand and explain poor inter-rater reliability ICC values, specifically for the DIS-CA. In a first step, absolute percentage agreement among the three raters was explored. In a second step, variability within region scores was analyzed, since it is well-known that a lack of variability of raw scores leads to lower ICC values [12]. Regarding absolute percentage agreement, no obvious pattern could be found explaining the poor ICC values, neither in the dystonia nor in the choreoathetosis subscale. Concerning variability, we did find differences in patterns that might be an explanation of poor ICC values. Analysis of variability for the DIS-CA revealed that, for all action items except eyes and trunk, more than half of the subjects were scored as a ‘0’ (i.e., having no choreoathetosis). This lack of variation (and not a low absolute agreement) might explain the poor inter-rater reliability values for the DIS-CA. Consequently, it is possible that the DIS is as reliable in inherited and idiopathic dystonia as in DCP regarding assessment of choreoathetosis.

In our study, dystonia was more frequently observed compared to choreoathetosis. Several participants had no choreoathetosis at all, which led to lower variability for DIS-CA compared to DIS-D. This may partly explain the generally higher DIS-D inter-rater reliability for region items compared to DIS-CA. For some DIS-D region items, poor inter-rater ICCs were found and in-depth analysis was performed to understand better. For the eyes, neck, arm right distal and leg right distal duration during action, more than half of the subjects were scored as a ‘4’, yielding a low variability and lower ICC values. For the trunk during action, the poor ICC values must be interpreted with caution, since half of the subjects were not scored, as they were not able to execute the requested actions.

Apart from the good to moderate inter-rater reliability, we found good to excellent test-retest reliability. For the total DIS, DIS-D, and DIS-CA, ICCs were 0.95, 0.88, and 0.93, respectively. These results are in agreement with Vanmechelen et al. [15], conforming that the DIS yields stable measures over time and thereby can be used for repeated assessments for individuals with both DCP and idiopathic or inherited dystonia. The values for body region items followed the same pattern as the body region items for inter-rater reliability. For the DIS-D, the eyes, neck, and arm right distal during action showed low variability within scores (half the subjects were scored with a ‘4’). For leg right proximal amplitude during action, more than half of the subjects were scored ‘0’ or could not perform the action, again leading to lower ICC values. For the rest of the values, the poor ICC values for arm right proximal, arm right distal and leg left distal duration can be explained by the high number (more than half) of the subjects that were scored as a ‘4’. For the DIS-CA, test-retest ICC values were very similar to the inter-rater ICC values, with lower ICC values explained by a lack of variability, as half of the subjects did not have choreoathetosis.

The SEM and MDD found in this study were 3.98% and 11.04%, respectively, for the total DIS. This means that a score difference of 11% is necessary to be sure that a ‘true’ improvement has occurred, rather than the difference being due to measurement error. For DIS-D, the SEM and MDD were 8.16% and 22.61%, respectively. The SEM and MDD for DIS and DIS-D in this study were lower than the values found for the BFM-M scale (SEM 9.88% and MDD 27.39%), which might yield more sensitive results when using the DIS to evaluate the effect of an intervention or disease progression [13].

Concurrent validity was investigated by correlation between the DIS-D and BFM-M scores, which was found to be good, r_s_ = 0.88. The BFM is currently the only recommended scale for assessing generalized inherited dystonia in adults [6]. To date, the BFM is often also used in children with dystonia (e.g., in intervention studies), since there is a lack of assessment scales designed for this population. This high correlation shows that the DIS measures are similar compared to the gold standard, the BFM. This evidence of concurrent validity suggests that both scales could be used to evaluate dystonia in future intervention studies in children and young adults with inherited and idiopathic dystonias.

In addition, due to the construct of the DIS, this scale can be used to describe the clinical patterns of dystonia and choreoathetosis in individuals with inherited or idiopathic dystonia, which has not been reported before. This might yield more accurate phenotypes compared to unstructured clinical observations. In-depth analysis of raw data in our study revealed that the presence of choreoathetosis was lower compared to dystonia. This is a new and clinically important finding, and the DIS could thus be used to map clinical patterns in inherited and/or acquired dystonias in future studies. These results can generate new insights on the distribution of both movement disorders, which may in turn help treatment delineation for each individual patient.

A limitation of this study is the rather small sample size. According to our power calculations, we chose to include 20 individuals. However, since some actions in the body region items could only be executed by half of the subjects, the ICC values of these actions should be interpreted with caution. With more participants included, these action items might have yielded more accurate results (smaller confidence intervals).

The DIS is the most comprehensive assessment tool currently available to evaluate the presence and severity of dystonia and choreoathetosis in a similar construct. For the individual being assessed, the time spent on filming according to the DIS video protocol is probably somewhat longer than in the case of the BFM-M. For the rater though, the DIS is much more time-consuming to score, since both dystonia and choreoathetosis are scored regarding amplitude and duration, in rest as well as in action. This makes the scale more feasible for scientific research rather than use in clinical practice. Effort is currently being made to explore statistical techniques that allow for a reduction of scoring options to shorten scoring time and to omit items that showed low agreement or many missing data. This may transform the DIS into a clinically attractive tool, which could lead to better consensus in assessing dystonia and choreoathetosis [10].

Another aspect to take into account is that the DIS yields a total percentage score. When evaluating interventions with the DIS, caution is thus warranted when comparing pre- and post-evaluation scores, as the number of performed actions might differ before and after the intervention. Only the items that were performed both pre- and post-intervention should be used in the comparative analysis and the items that were only present before or after the intervention should be discussed descriptively.

In conclusion, the DIS might be a promising tool to assess dystonia and choreoathetosis in children and young adults with inherited or idiopathic dystonia, since inter-rater reliability, test-retest reliability, and concurrent validity are shown to be good to excellent. This allows for more accurate evaluation of the effect in intervention studies, which may subsequently lead to better pharmacological or neurosurgical treatment management for populations with genetic and idiopathic dystonias. Furthermore, the natural course of these rare diseases might be better understood if individuals could be longitudinally evaluated.

## Figures and Tables

**Figure 1 jcm-09-02597-f001:**
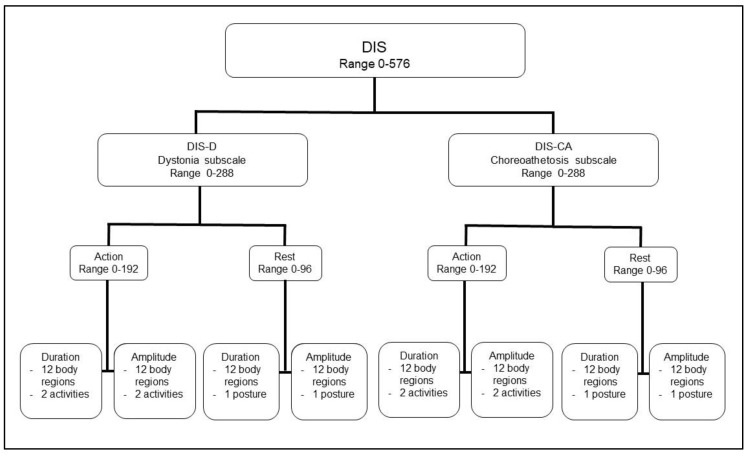
Diagram of the Dyskinesia Impairment Scale [5].

**Table 1 jcm-09-02597-t001:** Inter-rater reliability, intra-class correlation coefficient (ICC) values and confidence intervals (Cis) of the Dyskinesia Impairment Scale (DIS), *n* = 20.

	DIS Total Score, 0.83 (0.52–0.94)							
	**DIS–D, 0.87 (0.66–0.95)**				**DIS–CA, 0.71 (0.43–0.87)**			
**Body Regions**	**Action**		**Rest**		**Action**		**Rest**	
1 Eyes	0.46 (0.27–0.68)		0.70 (0.49–0.85)		0.44 (0.25–0.66)		0.57 (0.31–0.78)	
2 Mouth	0.68 (0.50–0.85)		0.60 (0.36–0.80)		0.71 (0.54–0.86)		0.82 (0.67–0.92)	
3 Neck	0.58 (0.38–0.80)		0.51 (0.20–0.74)		0.67 (0.47–0.85)		0.56 (0.30–0.77)	
4 Trunk	0.10 (−0.06–0.45)		0.40 (0.13–0.66)		0.79 (0.59–0.93)		0.69 (0.44–0.85)	
5 Arm R prox	0.80 (0.66–0.90)		0.55 (0.29–0.77)		0.47 (0.27–0.69)		0.66 (0.43–0.84)	
6 Arm L prox	0.64 (0.46–0.82)		0.71 (0.48–0.86)		0.50 (0.29–0.72)		0.62 (0.34–0.82)	
7 Arm R dist	0.61 (0.41–0.79)		0.70 (0.48–0.86)		0.65 (0.46–0.82)		0.79 (0.61–0.90)	
8 Arm L dist	0.67 (0.49–0.84)		0.70 (0.46–0.86)		0.59 (0.38–0.80)		0.61 (0.36–0.81)	
9 Leg R prox	0.45 (0.22–0.73)		0.78 (0.60–0.90)		0.20 (0.03–0.51)		0.46 (0.20–0.71)	
10 Leg L prox	0.64 (0.41–0.86)		0.78 (0.61–0.90)		0.20 (0.02–0.53)		0.58 (0.33–0.79)	
11 Leg R dist	0.58 (0.38–0.80)		0.59 (0.33–0.79)		0.37 (0.17–0.65)		0.56 (0.30–0.77)	
12 Leg L dist	0.71 (0.52–0.88)		0.65 (0.41–0.83)		0.27 (0.09–0.57)		0.44 (0.15–0.70)	
Total	0.85 (0.69–0.94)		0.82 (0.58–0.93)		0.61 (0.27–0.82)		0.79 (0.61–0.90)	
	**Action**		**Rest**		**Action**		**Rest**	
**Body regions**	**Duration**	**Amplitude**	**Duration**	**Amplitude**	**Duration**	**Amplitude**	**Duration**	**Amplitude**
1 Eyes	0.40 (0.22–0.63)	0.40 (0.22–0.63)	0.64 (0.41–0.82)	0.69 (0.46–0.85)	0.45 (0.26–0.67)	0.38 (0.20–0.61)	0.55 (0.29–0.77)	0.50 (0.24–0.74)
2 Mouth	0.64 (0.45–0.82)	0.62 (0.43–0.81)	0.59 (0.35–0.79)	0.51 (0.25–0.74)	0.68 (0.49–0.84)	0.69 (0.52–0.85)	0.87 (0.75–0.94)	0.62 (0.39–0.81)
3 Neck	0.47 (0.26–0.72)	0.62 (0.42–0.82)	0.38 (0.12–0.65)	0.55 (0.29–0.77)	0.59 (0.38–0.80)	0.66 (0.47–0.84)	0.62 (0.37–0.81)	0.34 (0.07–0.62)
4 Trunk	0.09 (−0.06–0.44)	0.05 (−0.08–0.37)	0.35 (0.09–0.63)	0.41 (0.14–0.68)	0.81 (0.62–0.94)	0.62 (0.37–0.86)	0.70 (0.46–0.86)	0.52 (0.26–0.74)
5 Arm R prox	0.79 (0.65–0.90)	0.72 (0.55–0.86)	0.34 (0.09–0.62)	0.69 (0.47–0.85)	0.44 (0.24–0.67)	0.44 (0.25–0.67)	0.67 (0.42–0.84)	0.61 (0.37–0.80)
6 Arm L prox	0.64 (0.46–0.82)	0.56 (0.37–0.76)	0.65 (0.41–0.83)	0.65 (0.41–0.83)	0.45 (0.24–0.69)	0.50 (0.30–0.72)	0.60 (0.33–0.80)	0.57 (0.31–0.78)
7 Arm R dist	0.51 (0.31–0.72)	0.63 (0.44–0.81)	0.60 (0.35–0.80)	0.67 (0.44–0.84)	0.61 (0.41–0.79)	0.66 (0.47–0.83)	0.79 (0.62–0.91)	0.70 (0.48–0.86)
8 Arm L dist	0.62 (0.43–0.81)	0.70 (0.52–0.86)	0.67 (0.43–0.84)	0.70 (0.45–0.86)	0.52 (0.32–0.75)	0.60 (0.39–0.80)	0.68 (0.45–0.85)	0.48 (0.20–0.73)
9 Leg R prox	0.46 (0.23–0.74)	0.29 (0.09–0.60)	0.69 (0.47–0.85)	0.76 (0.58–0.89)	0.14 (−0.01–0.44)	0.26 (0.07–0.58)	0.49 (0.22–0.72)	0.36 (0.10–0.63)
10 Leg L prox	0.64 (0.41–0.86)	0.51 (0.27–0.79)	0.71 (0.49–0.86)	0.78 (0.59–0.90)	0.19 (0.02–0.52)	0.15 (−0.02–0.48)	0.51 (0.25–0.74)	0.62 (0.38–0.81)
11 Leg R dist	0.49 (0.28–0.73)	0.62 (0.41–0.82)	0.49 (0.21–0.73)	0.63 (0.39–0.81)	0.28 (0.10–0.57)	0.46 (0.24–0.72)	0.72 (0.52–0.87)	0.34 (0.07–0.62)
12 Leg L dist	0.64 (0.44–0.84)	0.68 (0.48–0.86)	0.47 (0.19–0.72)	0.76 (0.56–0.89)	0.26 (0.08–0.56)	0.22 (0.05–0.52)	0.54 (0.27–0.77)	0.28 (−0.00–0.59)
Total	0.77 (0.59–0.89)	0.85 (0.68–0.94)	0.76 (0.48–0.90)	0.82 (0.64–0.92)	0.61 (0.24–0.82)	0.60 (0.32–0.81)	0.81 (0.65–0.92)	0.78 (0.60–0.90)

DIS-D = DIS dystonia subscale, DIS-CA = DIS choreoathetosis subscale, R = right, prox = proximal, L = left, dist = distal.

**Table 2 jcm-09-02597-t002:** Test-retest reliability, intraclass correlation coefficient (ICC) values and confidence intervals [CIs] of the DIS, *n* = 19.

	DIS Total Score, 0.95 (0.88–0.98)							
	**DIS–D, 0.88 (0.71–0.95)**				**DIS–CA, 0.93 (0.82–0.97)**			
**Body Regions**	**Action**		**Rest**		**Action**		**Rest**	
1 Eyes	0.46 (0.22–0.71)		0.58 (0.17–0.82)		0.59 (0.36–0.80)		0.58 (0.17–0.82)	
2 Mouth	0.81 (0.66–0.92)		0.75 (0.46–0.90)		0.63 (0.40–0.83)		0.75 (0.46–0.90)	
3 Neck	0.62 (0.38–0.82)		0.77 (0.50–0.91)		0.73 (0.53–0.88)		0.77 (0.50–0.91)	
4 Trunk	0.34 (0.05–0.71)		0.29 (−0.19–0.66)		0.73 (0.47–0.91)		0.29 (−0.19–0.66)	
5 Arm R prox	0.91 (0.83–0.96)		0.51 (0.07–0.78)		0.66 (0.45–0.83)		0.51 (0.07–0.78)	
6 Arm L prox	0.79 (0.63–0.90)		0.84 (0.63–0.93)		0.68 (0.47–0.84)		0.84 (0.63–0.93)	
7 Arm R dist	0.61 (0.39–0.80)		0.30 (−0.19–0.66)		0.56 (0.34–0.78)		0.30 (−0.19–0.66)	
8 Arm L dist	0.73 (0.51–0.89)		0.65 (0.27–0.85)		0.59 (0.34–0.81)		0.65 (0.27–0.85)	
9 Leg R prox	0.51 (0.22–0.78)		0.74 (0.44–0.89)		0.28 (0.04–0.61)		0.74 (0.44–0.89)	
10 Leg L prox	0.79 (0.59–0.92)		0.74 (0.38–0.90)		0.35 (0.08–0.67)		0.74 (0.38–0.90)	
11 Leg R dist	0.69 (0.46–0.87)		0.66 (0.30–0.86)		0.30 (0.05–0.62)		0.66 (0.30–0.86)	
12 Leg L dist	0.86 (0.71–0.95)		0.44 (−0.05–0.75)		0.37 (0.11–0.69)		0.44 (−0.05–0.75)	
Total	0.88 (0.71–0.95)		0.79 (0.54–0.92)		0.91 (0.78–0.96)		0.86 (0.63–0.95)	
	**Action**		**Rest**		**Action**		**Rest**	
**Body regions**	**Duration**	**Amplitude**	**Duration**	**Amplitude**	**Duration**	**Amplitude**	**Duration**	**Amplitude**
1 Eyes	0.36 (0.12–0.63)	0.43 (0.20–0.69)	0.55 (0.14–0.80)	0.62 (0.24–0.84)	0.61 (0.38–0.81)	0.40 (0.15–0.67)	0.81 (0.56–0.92)	0.71 (0.38–0.88)
2 Mouth	0.79 (0.62–0.91)	0.73 (0.53–0.88)	0.65 (0.29–0.85)	0.75 (0.46–0.90)	0.57 (0.33–0.79)	0.65 (0.42–0.84)	0.81 (0.58–0.92)	0.85 (0.64–0.94)
3 Neck	0.56 (0.31–0.79)	0.59 (0.35–0.81)	0.71 (0.38–0.88)	0.78 (0.51–0.91)	0.66 (0.44–0.85)	0.66 (0.42–0.85)	0.84 (0.63–0.94)	0.73 (0.43–0.88)
4 Trunk	0.35 (0.07–0.71)	0.37 (0.05–0.74)	0.38 (−0.09–0.71)	0.22 (−0.25–0.61)	0.69 (0.42–0.90)	0.79 (0.55–0.93)	0.64 (0.29–0.85)	0.77 (0.49–0.91)
5 Arm R prox	0.86 (0.74–0.94)	0.85 (0.72–0.93)	0.35 (−0.13–0.69)	0.68 (0.33–0.86)	0.62 (0.40–0.81)	0.60 (0.39–0.80)	0.60 (0.21–0.83)	0.44 (−0.02–0.74)
6 Arm L prox	0.73 (0.55–0.87)	0.72 (0.54–0.87)	0.78 (0.52–0.91)	0.62 (0.26–0.83)	0.67 (0.47–0.84)	0.60 (0.37–0.80)	0.83 (0.62–0.93)	0.80 (0.56–0.92)
7 Arm R dist	0.55 (0.31–0.76)	0.60 (0.33–0.80)	0.23 (−0.26–0.62)	0.31 (−0.17–0.67)	0.58 (0.36–0.79)	0.52 (0.29–0.75)	0.36 (0.10–0.69)	0.48 (0.06–0.76)
8 Arm L dist	0.76 (0.57–0.90)	0.64 (0.40–0.85)	0.59 (0.17–0.82)	0.69 (0.33–0.87)	0.54 (0.29–0.79)	0.58 (0.32–0.81)	0.81 (0.56–0.92)	0.74 (0.42–0.89)
9 Leg R prox	0.63 (0.36–0.85)	0.30 (0.07–0.62)	0.67 (0.32–0.86)	0.67 (0.33–0.86)	0.27 (0.03–0.60)	0.28 (0.04–0.62)	−0.07 (−0.53–0.41)	0.26 (−0.24–0.64)
10 Leg L prox	0.82 (0.65–0.93)	0.65 (0.40–0.85)	0.81 (0.57–0.92)	0.49 (0.08–0.77)	0.31 (0.05–0.64)	0.37 (0.10–0.69)	0.16 (−0.34–0.57)	0.15 (−0.34–0.56)
11 Leg R dist	0.65 (0.41–0.85)	0.63 (0.39–0.84)	0.61 (0.23–0.84)	0.50 (0.08–0.77)	0.28 (0.04–0.60)	0.24 (0.00–0.58)	0.50 (0.07–0.78)	0.39 (−0.08–0.72)
12 Leg L dist	0.85 (0.70–0.94)	0.76 (0.54–0.91)	0.34 (−0.16–0.69)	0.53 (0.09–0.80)	0.38 (0.11–0.70)	0.33 (0.08–0.66)	0.26 (−0.21–0.63)	0.36 (−0.11–0.70)
Total	0.90 (0.75–0.96)	0.92 (0.81–0.97)	0.76 (0.49–0.90)	0.77 (0.49–0.91)	0.83 (0.62–0.93)	0.87 (0.69–0.95)	0.83 (0.62– 0.93)	0.93 (0.83–0.97)

DIS-D = DIS dystonia subscale, DIS-CA = DIS choreoathetosis subscale, R = right, prox = proximal, L = left, dist = distal.

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
