# Peer review of "Reliability and Validity of the Dyskinesia Impairment Scale in Children and Young Adults with Inherited or Idiopathic Dystonia"

_jcm, 2020, doi:10.3390/jcm9082597_

Round 1
Reviewer 1 Report
Dear editor, I reviewed the article “Reliability and validity of the Dyskinesia Impairment Scale in children and young adults with inherited or idiopathic dystonia” by E Danielsson and colleagues
This work is of great interest and addresses an important clinical need for child neurologist and neurologist working with movement disorders. The article is well written, the methodology and the results are clearly presented. Tables 1 and 2 could not be seen completely.
The Discussion is clear and addresses the conflicting data (lower ICC for inter-rater reliability and test re-test reliability for body regions) and the limitations (low number of patients that could be fully scored with the DIS)
I have several comments to make:
About the study rationale: The authors had proven on a previous report that DIS was a reliable and valid scale for dyskinetic cerebral palsy. Was there any hypothesis regarding the possibility that validity and reliability of DIS would be different in other conditions associated with dystonia / choreoathetosis in children ?
The authors mention that they did not test for intra-rater reliability, although the high test re-test reliability was indicative of a high intra-rater liability. In my opinion formal assessment of intra-rater reliability would have been of interest to further assess the lower ICC’s for inter-rater reliability and test re-test reliability for body regions.
Given the lower ICC’s for parameters related to choreoathetosis, did the authors consider to test concurrent validity using some other scale ?
Was there any variability on the ICC for the different clinimetric parameters according to the GMFCS and the MACS ?
Author Response
Thank you very much for reviewing our manuscript! Your comments were very valuable to us. Please see the attachment for our reply.

Reviewer 2 Report
This study highlights the need for valid and reliable diagnostic and age range specific measures to assess the severity of different movement disorders in children and young adults. The results indicate that the DIS has the potential to be a useful outcome tool for children with inherited and idiopathic dystonia, a known gap in the assessment toolbox for this group. Greater accuracy in measuring dystonia and choreoathetosis pre and post medical interventions can only improve our understanding of the effectiveness of those interventions for different diagnostic groups. Thank you for starting the process of addressing this gap.
Comments below:
Introduction: This is very well written and informative.
On line 53 the authors indicate that several dystonia scales have been developed in the "past Decade" where in fact the DIS is the only scale that has been developed in the past 10 years, with the BADS (1999) and BFM (1985).
Participants: The power calculation indicates the need for 18 participants. This number was achieved however with so much missing data or data points that were unable to be collected, the actual number of participants with complete or near to complete data sets, is in fact much smaller. This is addressed on page 6 (Line 205-212) and again in the limitations but does make the reader question the strength of the outcomes.
3 participants are noted to be over the age of 22. As previous studies of the DIS have only included participants up to the age of 22, is this an issue?
Page 5: Scoring process - It is clear all 3 raters scored all first videos but video 2 was only scored by 1 rater?
Discussion:
On line 314 the authors report excellent test retest reliability yet have previously stated ICC's of 0.75 to 0.90 are good. The ICC for DIS-D was 0.88, which = good, not excellent as is reported.
Lines 326-330 discuss the SEM and MDD. Only the total DIS results are discussed. The SEM and MDD for the DIS-D was 8.16 and 22.61%, which I feel is the result most comparable to the BFM-M, yet is not discussed. I think this warrants further discussion.
Limitations: well outlined. It is a shame numbers were so low and that of the participants included complete data was not able to be gathered due to severity of the movement disorder.
I agree the DIS is the most comprehensive tool currently available but dont agree the video protocol is similar in length to the BFM-M. This may be the case for EM, author of the DIS, but not for other clinicians. The video process is very time consuming.
Conclusion: I agree the DIS appears a very promising tool for this diagnostic group and age range and may fill a gap in our assessment toolkit. It is exciting to hear that a more clinically friendly version of the tool is being considered as in its current form it is not a viable clinical tool. A study with larger numbers of participants, especially participants with choreoathetosis, is needed for the DIS to become an acceptable tool for use with this group of children and youth.
Final comment: It is really important to further understand and quantify movement disorders in children and young adults but this still leaves the question of how these movement disorders impact on their daily function and quality of life. We must continue to consider these more important aspects as a decrease in dystonia or choreoathetosis from an intervention does not necessarily translate to changes in function, independence or burden of care by others.
Author Response
Thank you very much for valuable comments! The manuscript has benefitted from the input. Please see the attachment for our reply.
